# Fabrication of Basalt Matrix Composite Material by Pressureless Aluminum Melt Infiltration in Air Atmosphere

**Roman A. Shishkin** [1,*], **Yuliy V. Yuferov** [2] **and Dmitriy O. Polyvoda** [3]

1. Institute of Solid-State Chemistry, Ural Branch of Russian Academy of Science, 620049 Yekaterinburg, Russia
2. The Department Chemical Engineering, Ariel University, Ariel 40700, Israel
3. Institute of High Temperature Electrochemistry, Ural Branch of Russian Academy of Science, 620049 Yekaterinburg, Russia
* Correspondence: shishkin@ihim.uran.ru; Tel.: +7-922-133-42-69

**Abstract:** The microstructure of Basalt matrix composite materials produced by pressureless aluminum melt infiltration at 950 °C was investigated. It is established that uniform elements distribution is observed within the whole sample depth. Interestingly, aluminum content variation considerably matches the hardness of the sample profile that is connected with alumina phase presence. Sample color changes during temperature treatment due to phase transitions were observed. The appearance of the hematite ($Fe_2O_3$) phase makes the initial preform red. After infiltration by molten aluminum, oxygen-deficient alumosilicate phases turn the color black. The infiltration process decreases the porosity insufficiently due to a partial reduction of alumosilicates by molten Al and the hardness of infiltrated samples was only 2.2 GPa. Nevertheless, a huge thermal conductivity rise from 1.45 to 4.53 W/(m·K), along with a fracture toughness increase, makes the produced composite a prospective wear-resistant material. Moreover, the developed low-temperature production technology allows for obtaining a very cost-effective material.

**Keywords:** basalt preform; composite material; infiltration; thermal conductivity; hardness

## 1. Introduction

The development of new wear-resistant materials to increase the operating time of parts under the aggressive influence of abrasive, erosive and cavitation loads is extremely important for various industries [1], e.g., for braking pads [2,3]. However, there is an important limitation in terms of the technical and economic feasibility of such materials' application, which leads to the rejection of expensive molding technologies (hot pressing, spark plasma sintering) and the need to reduce energy consumption, i.e., reducing the temperature, time and duration of sintering.

One of the methods of solving the problem is to obtain oxide-bonded silicon carbide at temperatures of 1400 °C [1,4]. Nevertheless, the obvious disadvantage of such materials is an increased porosity, which is why numerous publications are devoted to the impregnation of SiC preform with a melt of metallic aluminum or its alloys [5–8]. An alternative, cheaper but also sometimes less effective solution can be considering the use of basalt glass-ceramics [9–11]. Depending on the iron oxide content, the melting point of the mineral ranges from 1300 to 1500 °C [12–15]. This leads to the need to use expensive thermal equipment, which is a clear negative aspect for the industry and leads to the need to develop more affordable alternatives.

The basalt is sintered in the temperature range of 1000–1100 °C [16,17], which makes sintered basalt ceramics a cost-effective candidate as a wear-resistant material for industry. However, the sintering of cold uniaxially pressed samples results in considerable porosity of the produced materials [18]. Moreover, basalt has a low thermal conductivity [19,20], which in combination with high residual porosity can lead to overheating during the

tribological use of such materials [21–23]. These problems, as for silicon carbide [24], can be solved by infiltration with metallic aluminum melt under flux in air, thereby significantly reducing the requirements for the manufacturing design of the process. This approach will allow both for reducing the porosity of the composite material with an improvement in its operational properties, and for increasing thermal conductivity due to the use of metallic aluminum. Thus, this work is devoted to the study of the microstructure, and mechanical and thermal property changes of the porous basalt preform infiltration by an aluminum melt under flux in the air atmosphere.

## 2. Materials and Methods

### 2.1. Materials and Sample Preparation

Natural basalt rock and aluminum ingot (purity < 99.99%) were obtained from a local supplier. The rock was ground in the alumina ball mill. The resulting basalt powder (3.7 μm) was cold pressed at a pressure of 15 MPa and sintered in air atmosphere for 5 h at 1000 °C to produce the porous preform. The heating and cooling rate was set to 5 °C/min. Basalt glass ceramics were produced by primary crystallization process: natural ore was melted at 1300° and slowly (5 °C/min) cooled.

The basalt preform, with a density of 2.21 g/cc, thickness of 3.2 mm and diameter 22.2 mm, was placed into preheated aluminum melt under the KCl-NaCl (50–50 mol. %) flux in the alumina crucible at the temperature of 950 °C during 1 h and then extracted. The cooling of composite material was held in an air atmosphere. The Bt/Al composite material sintered tablet as well as natural and melted basalt samples were cut in a half by diamond saw and embedded in the epoxy resin (Epoxy 520). For the following study, the sample was polished at the automatic machine to a surface roughness less than 1.0 μm.

### 2.2. Characterization

The phase composition and structural studies were conducted using a Shimadzu XRD 7000 (Shimadzu Corporation, Japan) diffractometer ($Cu_{K\alpha}$ radiation ($\lambda$ = 1.5418 Å) between angles from 10 to 80 °, with a step of 0.03° and a shutter speed of 5 s at each point. Diffraction patterns were collected from the polished surface of the sintered tablet form sample.

The scanning electron microscopy (SEM) images were obtained from JEOL JSM 6390LA. Chemical composition of obtained materials was performed by Energy-dispersive X-ray spectroscopy (EDX) using Jeol JED2300 EDX-analyzer of JEOL JSM 6390LA. According to the SEM image, the porosity of the obtained sintered samples was calculated using ImageJ software 1.53t [25,26].

The Vickers hardness numbers were determined by a Zwick ZHU250 Universal Tester with a load of 10 kg for 15 s (HV10) according to ASTM E 384.

The fracture toughness coefficient ($K_{1c}$, MPa·m$^{1/2}$) was calculated by the following equation that showed adequacy in calculating the coefficient on the example of oxide and composite materials [27,28]:

$$K_{1c} = 0.16 H_V \cdot \sqrt{a} \cdot \left(\frac{c}{a}\right)^{-\frac{3}{2}} \tag{1}$$

where $H_v$—hardness (GPa); a—the diagonal length of the indenter (m); c—crack length (m).

Thermal conductivity investigation in temperature range of 25–150 °C was carried out at the custom-made installation IT-$\lambda$-400 discussed in detail earlier [29]. The measured value was adjusted for the porosity value according to the known equation [30]:

$$\frac{\lambda_{exp}}{\lambda_{dense}} = 1 - \frac{4}{3} \cdot P \tag{2}$$

where $\lambda_{exp}$ and $\lambda_{dense}$ are experimental and porous free sample thermal conductivity; W/(m·K); P—porosity of the specimen.

The measurement of density and open porosity was described in detail earlier [31].

## 3. Results and Discussion

The powder of ground basalt has a light grey color and presents a mixture of the following minerals (Figure 1): 61.8% anorthite ($CaSi_2Al_2O_8$), 17.3% tremolite ($Si_8Mg_5Ca_2O_{24}$), 11.2% chamosite ($Mg_{2.52}Fe_{2.48}Al_{1.2}Si_{3.8}O_{18}$), 8.5% corundum ($Al_2O_3$) and minor traces, which could be identified as about 1.2% of chlorite ($Mg_{9.17}Fe_{1.02}Al_{3.46}Si_{6.35}O_{36}$). After sintering, the sample color changed to a brick red, which could be connected with a high $Fe^{3+}$ content in the natural basalt. Sintering was accompanied by linear shrinkage of 11.2%. The sample XRD pattern (Figure 1) clearly reveals a two-phase formation: 24.1% ferrosilite ($Si_{16}Mg_{5.09}Fe_{10.65}Ca_{0.26}O_{48}$) and 5.8% hematite ($Fe_2O_3$). The element content calculation from the XRD refinement results shows no evident difference in Si ($20 \pm 1\%$) and Al ($12.0 \pm 0.6\%$) amounts after the sintering of basalt. At the same time, calcium and magnesium content drops from 12.9 and 3.8 to 8.7 and 2.2% correspondingly and the iron amount rises from almost 0 to 9%. It is likely that most parts of the iron exists in $Fe^{2+}$ form in the natural basalt mineral and partially substitute Ca and Mg in anorthite, tremolite and chamosite. During the sintering process, iron oxidizes to the 3+ form with an effective atomic radii reduction that forces the formation of $Fe^{3+}$ phases such as hematite, accompanied by the acquisition of the color red in the specimen.

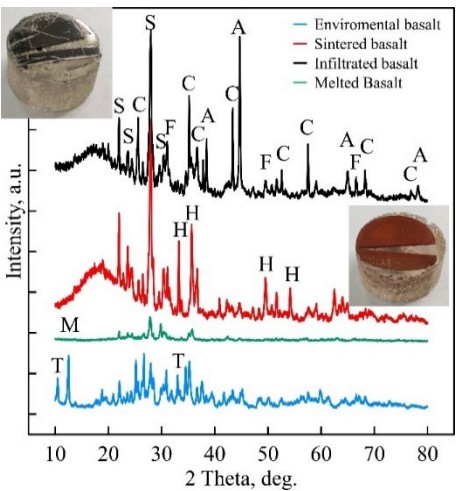

**Figure 1.** XRD patterns and images of basalt samples: A—aluminum (COD 96-431-3215); C—corundum (COD 96-900-9681); F—Ferrosilite (COD 96-900-0360); S—Anorthite (COD 96-900-0363); H—hematite (COD 96-901-5965); T—tremolite (COD 96-900-0360); M—chamosite (COD 96-900-9234).

After the infiltration process, the sample color changed to black, which could be the result of oxygen non-stoichiometric, which is going to be discussed later. Hematite absence is due to dissolution in $Al_2O_3$ [32]. Both metal aluminum (9.4%) and $\alpha$-$Al_2O_3$ (39.7%) peaks are clearly observed. The $\alpha$-$Al_2O_3$ forms during the cooling of the composite material in the air atmosphere, meanwhile anorthite and ferrosilite stay without considerable changes after the infiltration process.

An image of the natural basalt microstructure (Figure 2a) clearly illustrates four different phases, differing in density. The lightest phase (dark gray spots) is anorthite 2.73 g/cc, a mixture of binding phases is tremolite (2.98) and chamosite (3.13), and bright white corundum inclusions (4.01 g/cc). The fused basalt consists of two phases with significantly different densities (anorthite and pigeonite ($Mg_{4.88}Fe_{3.12}Si_8O_{24}$)) 2.73 and 3.38 g/cc, respectively. However, as shown (Figure 2b), these phases are practically indistinguishable, despite 66.9% crystallinity, which is associated with the formation of a fine homogeneous-like microstructure with sufficiently rapid cooling of the melt.

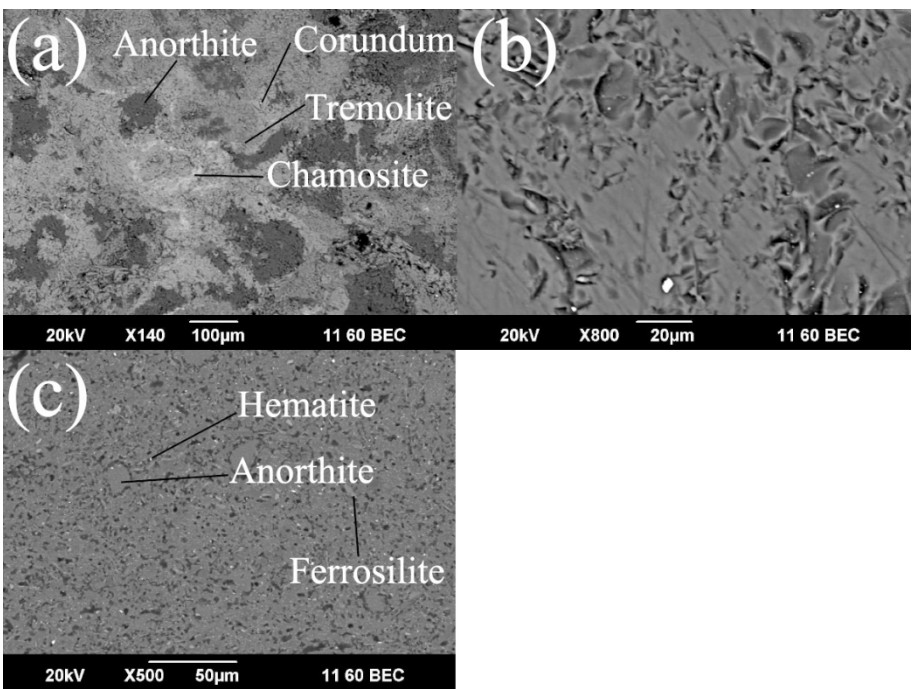

**Figure 2.** SEM images of (**a**) natural basalt ore; (**b**) basalt glass ceramics; (**c**) sintered basalt preform.

The SEM image (Figure 2c) shows a basalt preform polished surface with a moderate porosity of 12.6%, which is in a good agreement with experimental data of an open porosity of 10.5%. In the picture, three phases are clearly distinguishable, significantly differing in density. The main part is occupied by the anorthite phase with a density of 2.73 g/cc, on which inclusions of a heavier (lighter) phase in large quantities are clearly visible, which corresponds to the ferrosilite phase (3.52 g/cc). Small light spots are observed separately, which belong to the hematite phase, which fully corresponds to the data obtained when describing XRD patterns.

The composite material sample calculated porosity is about 8.8 ± 1.2%. Unlike the infiltrated oxide-bonded silicon carbide [24], the basalt, with almost complete porosity removal, is not observed in the sample's surface area. This effect differs in that in the case of SiC, the aluminum melt dissolves only the $SiO_2$ oxide film on the surface of the grains, while in basalt, aluminum is impregnated into the composition of the aluminosilicate, not accompanied by a significant change in volume. Thus, because of the absorption of aluminum in the aluminosilicate phase, small pores are formed after cooling, giving the above porosity.

The composite material matrix, as in the case of the preform, consists of the anorthite phase. Aluminum and anorthite are clearly distinguishable within SEM. Metallic aluminum is found exclusively in structural defects, such as cracks and pores. When directly in contact with silicates, aluminum enters their structure or forms an oxide phase as mentioned earlier, which contributes to a change in the color of the sample. The individual silicate grains also observed (Figure 3b) undoubtedly belong to the ferrosilite phase remaining after infiltration. There are clearly visible light dots, which is an $\alpha$-$Al_2O_3$ phase. Interestingly, numerous $\alpha$-$Al_2O_3$ growths were found in the ferrosilite phase, which may explain the blackening of the resulting material. It is known that non-stoichiometric oxygen compounds have a black color [33]. The only iron between all main elements found in basalt is a 3d transition metal that can form black-colored oxygen-deficient phases. Thus, molten aluminum at 950 °C can partially restore iron from $Fe^{3+}$ to $Fe^{2+}$, thereby binding part of the oxygen from the ferrosilite phase (3), coloring it black. According to thermodynamic calculations, aluminum is able to reduce iron to a metallic state. However, due to the low content of aluminum melt in the pores and the low rate of diffusion processes, the degree of such interaction

is negligible, which is confirmed by the absence of the intermetallic phase FeAl$_3$ or pure metallic iron according to XRD and SEM.

$$Si_{16}Mg_{5.09}Fe_{10.65}Ca_{0.26}O_{48} + n\ Al^{liq} = Si_{16}Mg_{5.09}Fe_{10.65}Ca_{0.26}O_{48-1.5n} + \frac{n}{2}Al_2O_3 \quad (3)$$

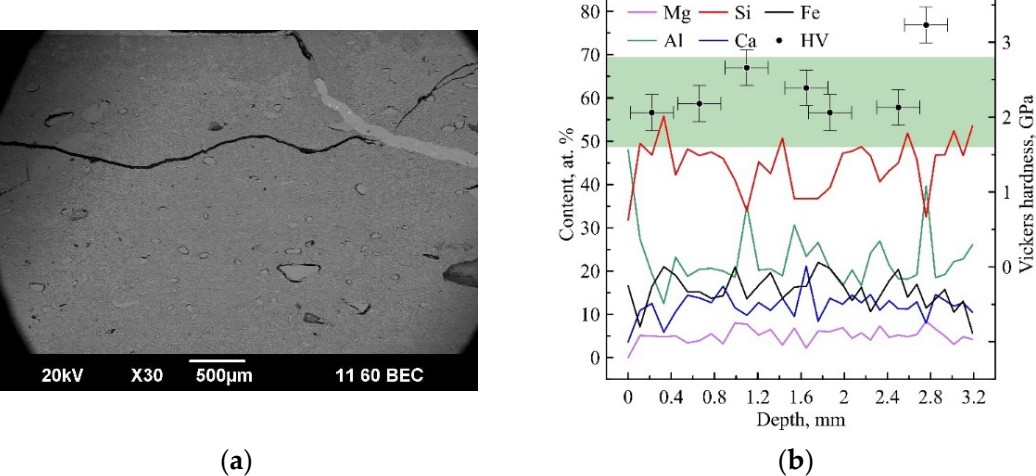

| (a) | (b) |

**Figure 3.** SEM image (**a**) Al-zone distribution, (**b**) distribution of elements on sample.

Figure 3 shows the distribution of basic elements of basalt, namely Al, Mg, Si, Ca, and Fe content in the infiltrated sample. It should be noted that the elemental content is in good agreement with the XRD data discussed earlier. That was estimated by EDX-profiling on the SEM, only for elements Al, Mg, Si, and Ca, while C and O were not estimated due to the lack of calibration of the EDX-spectrometer on light elements and the presence of vapor vacuum oil in the chamber of SEM.

The hardness investigation (Figure 3b) showed a small deterioration of the indicator for the composite material (2.2 GPa) compared to the initial preform (2.7 Gpa), while in the case of silicon carbide, an increase in hardness is observed in the infiltrated layers [24]. This effect is associated with a significant decrease in the porosity of the SiC preform during infiltration by aluminum melt, while in the case of basalt, porosity decreases by only 3% (from 11.9 to 8.8%). Additionally, a partial reduction of iron by aluminum melt leads to a decrease in hardness, as well as the hardness of aluminum itself (~0.4 GPa [31]). At the same time, the hardness distribution over the depth of the sample does not have a pronounced gradient character, which confirms the uniformity of aluminum infiltration. There is a clear dependence of the composite material hardness on the aluminum content (Figure 3b), which is associated with increased content of the corundum phase ($\alpha$-Al$_2$O$_3$), which has a high hardness.

With a hardness decrease, growth in the crack resistance of the composite material is observed. Due to the high fragility of natural and basalt glass ceramics, indenter traces are indistinguishable due to cracking and collapse of the imprint boundaries (Figure 4a,b). The long cracks are clearly visible and have dispersed along the sample (Figure 4b). The value of the crack resistance coefficient (K$_{1c}$) for a sintered basalt sample is 2.7, which exceeds the value of 2.2 MPa·m$^{1/2}$ obtained in [16]. In turn, for the aluminum-infiltrated composite material, K$_{1c}$ = 3.5 MPa·m$^{1/2}$, which is only slightly inferior to silicon carbide [34].

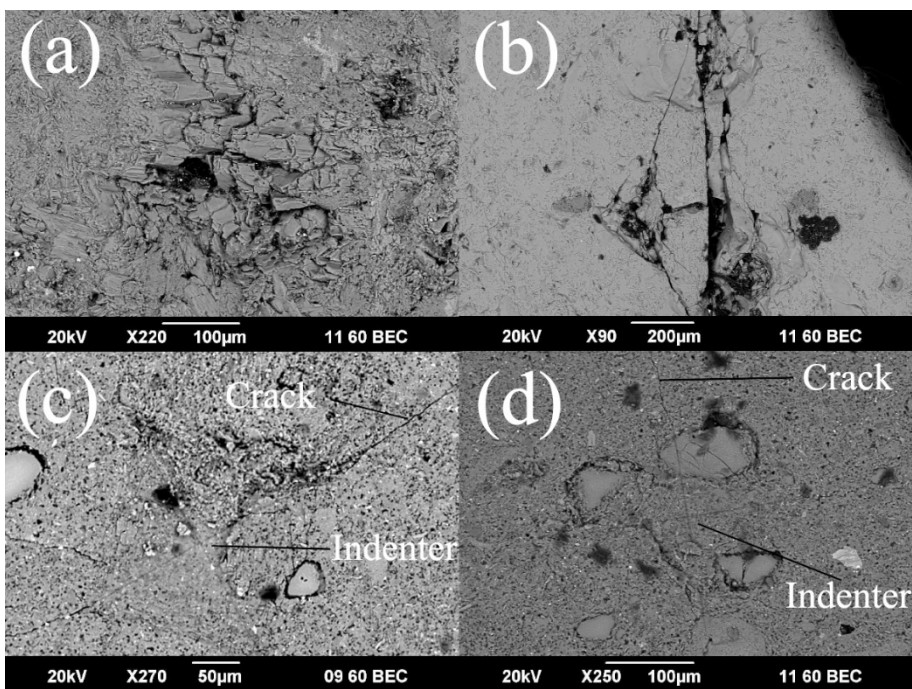

**Figure 4.** SEM images of Vickers indenter (**a**) natural basalt; (**b**) basalt glass ceramics; (**c**) sintered basalt; (**d**) composite material.

It was observed that the crack mainly spreads through the matrix of the composite material (Figure 5). The matrix consists of two highly dispersed alumosilicate phases (anorthite and ferrosilite), even at the high magnitude, the character of the cracking cannot be obviously supposed. It can be seen (Figure 5b) that a crack passes through both phases, for which the grain size is lower than the crack dimension.

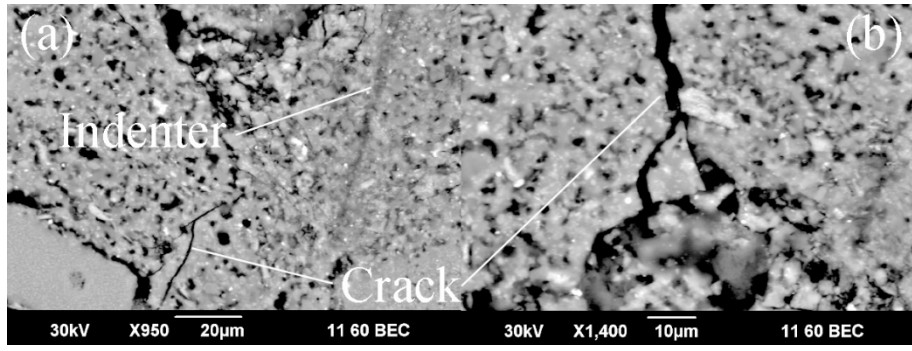

**Figure 5.** SEM images of crack after HV indenter: (**a**) 950×; (**b**) 1400×.

Due to the high fragility, we were unable to produce a sample of the required size from basalt glass ceramics. Therefore, the study of the thermal conductivity of basalt was carried out using the example of a sintered sample and a composite material (Figure 6). The highest accuracy of measuring thermal conductivity was obtained because a sample series with different heights of sintered basalt tablets (diameter 15 mm, height 3–5 mm) and composite material (diameter 15 mm, height 5–8 mm) was made. The given height values were selected based on the experimental standard for the custom-made apparatus. Due to the fixation of the temperature difference at the ends of a plane-parallel polished sample, there is an unambiguous correlation between the accuracy of measuring thermal conductivity and the thickness of the tablet. The lower the expected thermal conductivity, the lower the thickness of the test sample, and vice versa. The obtained values of the thermal conductivity of sintered basalt significantly exceed the data given in [19,20], but at the same time are

significantly inferior to the thermal conductivity of basalt rock [35]. Such an impressive difference in the literature data is explained by different deposits of basalt, as a consequence of different morphology and chemical composition, as well as different methods of thermal conductivity research. The method of measuring thermal conductivity proposed and tested by our group ensures the high reliability of the results [29]. Comparison of the thermal conductivity values of composite materials with basalt fibers [36,37], as well as with basalt rock [35], shows a significant superiority of the developed composite materials (4.53 ± 0.22 W/(m·K)). The thermal conductivity that increased by two times compared to the sintered sample will allow the material to remove the heat caused by abrasive tribological action more efficiently during operation.

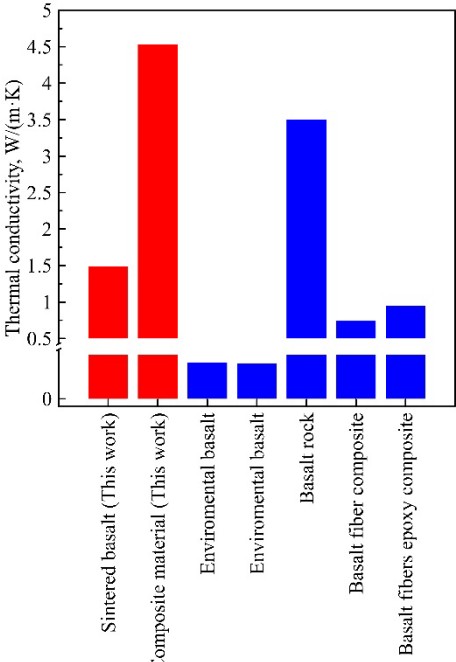

**Figure 6.** Thermal conductivity of experimental and literature data [19,20,35–37].

The combination of enhanced mechanical and thermophysical properties of the resulting composite material, together with cheap low-temperature production technology, can be beneficial in the development and manufacture of commercially available wear-resistant materials for industry, e.g., braking pads. Additionally in this work, for the first time, the method of infiltration by molten metallic aluminum to a porous preform obtained from natural basalt powder was applied.

## 4. Conclusions

The sintered basalt matrix composite material was produced by the pressureless aluminum melt infiltration in an air atmosphere at 950 °C. Within the sintering of basalt powder, $Fe^{2+}$ to $Fe^{3+}$ is oxidized, which leads to a change of color to red. However, after infiltration with metallic aluminum, a partial reduction of iron occurs with the formation of aluminosilicates with oxygen non-stoichiometry, which leads to the blackening of the sample. The main difference in the phase composition of sintered basalt and the composite material is the presence of a phase of hematite ($Fe_2O_3$) in the initial preform, which dissolves in the resulting aluminum oxide and other aluminosilicates. EDX depth mapping of the sample indicates uniform infiltration by aluminum and fluctuations in the aluminum content (associated with areas enriched with aluminum oxide) clearly correlate with changes in hardness throughout the depth of the sample.

After the infiltration process, there is no significant change in the microstructure of the sample. However, residual porosity remains due to the reaction of metallic aluminum

and aluminosilicates during the impregnation process. The resulting composite material has a similar hardness to the original preform, but significantly greater crack resistance, which is caused by a decrease in porosity and a developed multiphase microstructure. Moreover, aluminum impregnation leads to a significant increase in thermal conductivity up to 4.53 W/(m·K), which, together with increased mechanical properties and simple low-temperature production technology, makes the developed materials attractive for use as wear-resistant materials, such as brake pads.

**Author Contributions:** Conceptualization, R.A.S.; methodology, R.A.S. and Y.V.Y.; validation, R.A.S. and Y.V.Y.; formal analysis, R.A.S. and Y.V.Y.; investigation, R.A.S., Y.V.Y. and D.O.P.; resources, R.A.S.; data curation, R.A.S. and Y.V.Y.; writing—original draft preparation, R.A.S.; writing—review and editing, R.A.S. and Y.V.Y.; visualization, R.A.S.; supervision, R.A.S.; project administration, R.A.S. All authors have read and agreed to the published version of the manuscript.

**Funding:** This work was supported by state assignment No. AAAA-A19-119110190048-7.

**Institutional Review Board Statement:** Not applicable.

**Informed Consent Statement:** Not applicable.

**Data Availability Statement:** Not applicable.

**Conflicts of Interest:** The authors declare no conflict of interest.

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
