# Peer review of "Fabrication of Basalt Matrix Composite Material by Pressureless Aluminum Melt Infiltration in Air Atmosphere"

_ceramics, doi:10.3390/ceramics5040056_

Round 1
Reviewer 1 Report
This paper contanins some very interesting redults. and it's qualified for publication. Some suggestions are listed as following:
1. The different phases are suggested to be indicated in Figure 2 to improve readability.
2. I suggest the authors to indicate cracks in figure 4.
Author Response
Dear reviewer, thank you for your work and evaluation of the submitted work! We have carefully studied and analyzed your suggestions and made the appropriate edits, namely:
- The phases were identified at the figure 2.
- Both cracks and indentor inprint were indicated at the figure 4.

Reviewer 2 Report
Dear Authors,
The Abstract section is successful to clear the rest of your article.
1-in the methodology section, why the authors choose this formula for fracture toughness calculation? There are numerous formulas based on the crack length and diagonal to length ratio to select the most appropriate fracture toughness equation.
2-in the results and discussions section, the same format should be followed throughout the text. such as, Fig.1 (not fig.1),or Figure 1; Figure 2a or Fig. 2a ?
either alumina or Al2O3, g/cm3 instead of g/cc, either Al or Aluminum
3-in line 163, what is the 3d metal?
4- Figure 4, SEM micrographs should be clearer and clearly show the crack paths. Either the crack transgranular or intergranular? Use higher magnification of SEM micrographs.
5- over citation of authors own published articles is also detected.
Author Response
Dear reviewer, thank you for your work and evaluation of the submitted work! We have carefully studied and analyzed your suggestions and made the appropriate edits, namely:
- The additional reference was added to the k1c formula and the description added. The similar oxide and metal infiltrated composite materials were studied with the equation in the mentioned reference.
- The same format was applied for figures, units and chemical formula.
- “3d transition metal” was added in the 163 line.
- The new SEM images with a higher both magnification and voltage were added in the figure 5.
- 1 reference to the paper of our group was excluded. The rest 3 references are necessary: the closest infiltration process by aluminum in air was carried by our group. The thermal conductivity apparatus also is only described only in the reference was given. Some experimental procedures as well as hardness values obtained earlier should be linked.
